# Receptor for the Advanced Glycation End Products (*RAGE*) Pathway in Adipose Tissue Metabolism

**DOI:** 10.3390/ijms241310982

**Published:** 2023-07-01

**Authors:** Klaudia Gutowska, Krzysztof Czajkowski, Alina Kuryłowicz

**Affiliations:** 1II Faculty and Clinic of Obstetrics and Gynaecology, Medical University of Warsaw, 00-315 Warsaw, Poland; klaudia.gutowska@wum.edu.pl (K.G.); krzysztof.czajkowski@wum.edu.pl (K.C.); 2Doctoral School, Medical University of Warsaw, Zwirki i Wigury 81, 02-091 Warsaw, Poland; 3Department of Human Epigenetics, Mossakowski Medical Research Centre PAS, 02-106 Warsaw, Poland; 4Department of General Medicine and Geriatric Cardiology, Medical Centre of Postgraduate Education, 00-401 Warsaw, Poland

**Keywords:** advanced glycation end products (AGEs), receptor for AGE (*RAGE*), obesity, adipose tissue dysfunction, metabolic inflammation, insulin resistance, diabetes, cardiovascular diseases

## Abstract

Advanced glycation end products (AGEs) are mediators in the process of cellular dysfunction in response to hyperglycemia. Numerous data indicate that the accumulation of AGEs in the extracellular matrix plays a key role in the development of obesity-related adipose tissue dysfunction. Through binding of their membrane receptor (*RAGE*), AGEs affect numerous intracellular pathways and impair adipocyte differentiation, metabolism, and secretory activity. Therefore, inhibiting the production and accumulation of AGEs, as well as interfering with the metabolic pathways they activate, may be a promising therapeutic strategy for restoring normal adipose tissue function and, thus, combating obesity-related comorbidities. This narrative review summarizes data on the involvement of the *RAGE* pathway in adipose tissue dysfunction in obesity and the development of its metabolic complications. The paper begins with a brief review of AGE synthesis and the *RAGE* signaling pathway. The effect of the *RAGE* pathway on adipose tissue development and activity is then presented. Next, data from animal and human studies on the involvement of the *RAGE* pathway in obesity, diabetes, and cardiovascular diseases are summarized. Finally, therapeutic perspectives based on interference with the *RAGE* pathway are discussed.

## 1. Introduction

Over the years, the concept of the role played by adipose tissue in maintaining human homeostasis has evolved. Initially treated solely as a mechanical insulator and energy store, it is now seen as the source of a number of substances and mediators (adipokines) that have the ability to regulate the function of organs and tissues throughout the body [1]. Consequently, the dysfunction of adipose tissue impacts the homeostasis of the entire body. This situation takes place in the course of obesity, when excess nutrients accumulated in adipocytes lead to mitochondrial dysfunction and associated oxidative stress [2]. In turn, oxidative stress can increase preadipocyte proliferation, adipocyte differentiation, and the size of mature adipocytes. In obesity, oxidative stress is not limited to adipose, and reactive oxygen species (ROS) can impair the function of hypothalamic neurons that control satiety and hunger behavior in a vicious circle mechanism [3].

These phenomena lead to reprogramming of the profile of genes expressed in the adipocyte and thus to a change in the adipokines it secretes. This process, termed adipose tissue dysfunction, is thought to underlie the development of insulin resistance predisposing to glucose intolerance and a range of other chronic complications associated with obesity, affecting virtually all organs in the human body and significantly impairing quality of life [4].

However, adipose tissue dysfunction is not just about adipocyte metabolism. Changes that occur in the adipose tissue extracellular matrix (ECM) play an equally important role in this process. The ECM is a complex structure composed of various proteins, polysaccharides, and proteoglycans and constitutes a scaffold for cells to modulate their biological processes [5]. Obesity is characterized by the massive expansion of adipose tissue; therefore, remodeling and reorganization of the ECM are necessary to provide sufficient space for adipocyte enlargement (hypertrophy) and for the formation of new adipocytes through adipogenesis from precursor cells (hyperplasia). An important component of normal adipose tissue expansion is the formation of new blood vessels, as this disruption results in adipocyte necrosis, as well as hypoxia, which trigger chronic low-grade inflammation and fibrosis, exacerbating adipose tissue dysfunction and leading to increased insulin resistance [6].

Advanced glycation end products (AGEs), resulting from the non-enzymatic glycation of lipids and proteins, are mediators in the process of cell dysfunction in response to hyperglycemia. AGEs disrupt the tertiary structure of proteins and alter their function and structural interactions. In addition, acting as ligands for scavenger receptors present on many cells (such as CD36 or the receptor for advanced glycation end products: *RAGE*), they affect numerous intracellular pathways [7,8]. In the course of hyperglycemia (both in diabetes and prediabetes), the concentrations of AGEs increase in several locations, including the ECM of adipose tissue [9]. Preclinical and clinical studies indicate that the accumulation of AGEs in the ECM may interfere with the proper differentiation of adipocyte precursors [10,11] and affect the metabolism of mature cells, including their lipid sto*RAGE* capacity [12,13], insulin sensitivity [13,14,15], secretory activity [16], and production of inflammatory mediators [17,18]. Therefore, inhibiting the production and accumulation of AGEs, as well as interfering with the metabolic pathways they activate, may be a promising therapeutic strategy for restoring the normal function of adipose tissue and thus in combatting obesity-related comorbidities.

The aim of this narrative review is to summarize the data on the role of the *RAGE* pathway in adipose tissue dysfunction in obesity and its metabolic complications. First, the mechanisms of AGE formation and the *RAGE* signaling pathway are briefly presented. Next, we outline the role of the *RAGE* pathway in the regulation of adipogenesis and adipose tissue function. We then summarize data from animal and human studies on the involvement of the *RAGE* pathway in obesity, diabetes, and cardiovascular disease. Finally, we discuss therapeutic perspectives based on interference with the *RAGE* pathway.

## 2. Advanced Glycation End Products (AGEs) and the Receptor for Advanced Glycation End Products (RAGE) Pathway

### 2.1. Advanced Glycation End Products

AGEs are a heterogeneous group of compounds formed exogenously or endogenously from different precursors and by different mechanisms. One criterion for the division of AGEs is their origin: compounds absorbed from food or cigarette smoke are called exogenous AGEs, and those produced in tissues and body fluids are endogenous; these, in turn, can be further divided according to their precursor. High temperatures and extended food preparation times increase the formation of these molecules. Notably, exogenous AGEs are formed more rapidly and in larger quantities. Regardless of the source, AGEs can also be divided into cross-linkers, non-fluorescent cross-linker products, or non-fluorescent non-cross-linkers. However, the exact categorization of AGEs by structure has been described elsewhere and is not the subject of this article [19,20,21]. To date, more than 20 different AGEs have been characterized, including, for example, lysine carboxymethyl (CML), lysine carboxyethyl (CEL), pentosidines, and lysine methylglyoxal dimers. These compounds have received increased research attention due to their role in the development of chronic conditions such as diabetes, cardiovascular and neurological diseases, and cancer [19,22].

AGEs present in adipose tissue can be either exogenous (supplied by the circulation) or endogenous (formed locally as a result of adipose-tissue-dysfunction-related oxidative stress or diabetes-related hyperglycemia). Hyperglycemia not only promotes formation the of endogenous AGEs but also their accumulation. Glycation (i.e., the non-enzymatic glycosylation of amino residues) is a spontaneous, two-stage process, the intensity of which depends on the content of simple sugars, including glucose, in the body. First, unstable Schiff bases are formed through the direct correlation between the carbonyl groups of reducing sugars and the free amino groups of nucleic acids, proteins, or lipids. If amino acids are included, this can lead to disturbances in the structure and function of proteins crucial for cells’ organization and metabolism [19,20,21]. In the case of nucleic acids, the glycation of amino residues promotes the breaking of nucleotide chains and the formation of mutations, while the glycation of cell membrane lipids (e.g., Phosphatidylethanolamine) contributes to their increased permeability. Second, molecules known as Amadori products may be modified in two ways: converted by oxidation or hydrolysis into AGEs or by forming AGE precursor compounds, such as glyoxal (GO) and methylglyoxal (MGO). In addition to Amadori products, a heterogenous class of reactive carbonyl is created and can be converted into protein adducts or protein crosslinks. As a heterogenous group, AGEs are also generated through the Maillard pathway, the Wolff pathway (oxidation of monosaccharides), and the Namiki pathway (amino acid or lipid degradation and the cleavage of dicarbonyl compounds from aldimines) [19,21]. AGE formation depends on the concentration and reactivity of glucose, the amount of AGE precursors, and the availability of free amino groups. The Wolff (polyol) pathway is active under hyperglycemic conditions and involves the formation of sorbitol from glucose. Excessive activation of this pathway leads to increased dicarbonyl formation through the accumulation of upstream metabolites such as fructose. Endogenous lipid peroxidation, resulting from the ROS-induced lipid peroxidation of polyunsaturated fatty acids present in cell membranes, can also lead to the increased formation of dicarbonyls, and subsequently, AGEs (the Namiki pathway). AGEs formed by reactive dicarbonyls produced by lipid peroxidation are also referred to as ALEs (advanced lipoxidation end products) [23].

### 2.2. Receptor for the Advanced Glycation End Products (RAGE) Pathway

Both endogenous and exogenous AGEs have the ability to bind to a specific *RAGE* receptor present on the surface of, among others, endothelial cells, muscle cells, immunocompetent cells, glomerular podocytes, and neurons. *RAGE* expression has been found in many adipose-tissue-forming cells: adipocytes, their precursors, stromal cells, vasculature, or infiltrating macrophages [24].

The binding of AGEs to a specific receptor of the immunoglobulin superfamily becomes a signal for the intracellular formation of reactive oxygen derivatives and the activation of transcription factors. As a result of the oxidative stress developing in cells, the pro-inflammatory nuclear factor κB (NF-κB) is activated, as well as other signaling pathways (including, for instance, mitogen-activated protein kinases (MAPK), NADPH oxidase, extracellular regulated(ERK)-1/2, and c-JunN-terminal kinase (JNK)), leading to the increased transcription of genes encoding cytokines and adhesion molecules [19,22,25,26]. The inflammatory process that develops in this way promotes organ damage and the development of chronic complications associated with both aging and obesity. *RAGE* has been shown to engage multiple diverse ligands, not only AGEs but also damage-associated molecular pattern molecules, such as high-mobility group box 1 (HMGB1) S100 proteins and amyloid-β peptide [27,28]. The pathways activated upon *RAGE* binding are often cell- and ligand-specific. For instance, in murine macrophages (RAW cells), the stimulation of *RAGE* by glycated bovine serum albumin (BSA) leads to the activation of the Jak–Stat pathway, while the binding of *RAGE* with CML in rat mesangial cells results in the activation of angiotensin II signaling [23]. A simplified scheme of *RAGE*-mediated signaling is shown in Figure 1.

Several studies have supported the hypothesis that, in addition to the full-length form at the cell surface, *RAGE* also exists in other non-membrane-bound forms. Soluble *RAGE* (s*RAGE*) is produced by either the proteolytic cleavage of the full-length form of *RAGE* (c*RAGE*) or alternative mRNA splicing and lacks trans-membrane and cytoplasmic domains. Importantly, s*RAGE* can inhibit full-length *RAGE* activation by binding to their ligands in the extracellular space [29].

## 3. Influence of the RAGE Pathway on Adipose Tissue Activity

Both white (WAT) and brown adipose tissue (BAT) are flexible organs that play a prominent role in regulating energy homeostasis [30]. In addition, adipocytes have been proven, inter alia, to defend internal organs from mechanical harm and secrete biologically active substances capable, in an endocrine manner, of controlling the function of several organs and tissues throughout the body [31]. Most recently, the relationship between *RAGE* and adipose tissue activity has gained more attention due to the possible modulation of adipocytes’ functioning.

### 3.1. Influence of the RAGE Pathway on Adipogenesis and Adipose Tissue Browning

Adipose-derived stem cells (ASCs) are multipotent cells naturally occurring in adipose tissue, and which are able to differentiate into adipogenic, osteogenic, chondrogenic, and other types of cells. Although ASCs are mostly derived from white adipose tissue, they are also present in BAT but with different characteristics [32,33].

Exposure to a high-fat, high-sugar diet negatively impacts on ASCs in adipose and bone tissue. Excessive energy intake promotes AGE formation, which impairs the proliferation and differentiation potential of ASCs, manifested as the decreased cell counting kit-8 (CCK-8) protein level and alkaline phosphatase (ALP) activity. However, the effect can be species specific. In mice, a high-fat and high-carbohydrate diet leads to a lower expression of osteogenic differentiation genes (Alp, Opn, Ocn, and Runx2) in ACCs. This effect may result from DNA methylation in ASCs and the key role this process plays in the Wnt/β-catenin signaling pathway. Indeed, AGEs drive the downregulation of Wnt signaling molecules (β-catenin, lymphoid enhancer-binding factor 1 (LEF1), and glycogen synthase kinase 3 beta GSK3β), while enhancing the expression of methyltransferase genes [34]. However, in vitro studies using human mesenchymal stem cells (MSCs) suggest that exposure to AGEs inhibits their adipogenic differentiation (assayed by oil red O staining, lipoprotein lipase production, and intracellular triglyceride content) without incurring significant impairments in osteogenic development [10].

In addition to the evidence for AGE–*RAGE* pathway involvement in the regulation of ASCs’ fate, there are data on its contribution to adipose tissue expansion via the modulation of adipocyte senescence. For instance, in murine preadipocytes (3T3-L1 cells), activation of the AGE–*RAGE* axis, probably by the blockade of the p53 protein, restores the adipogenic potential of these cells. Interestingly under these experimental conditions, AGEs showed no effect on adipogenesis in young preadipocytes, and the exact mechanism of this phenomenon remains unknown [35].

Regardless of the direct impact on adipocytes, AGE formation in hyperglycemic conditions damages other components of adipose tissue. Recent studies on 3T3-L1 preadipocytes suggest that the accumulation of AGEs in the ECM, independently of the *RAGE* pathway, impairs adipocyte differentiation [36]. In addition, through ECM modification, AGEs have direct effects on cell niches and plasma membranes via loss of their plasticity and therefore induce alterations in cellular signaling and cytoskeletal organization. Collectively, with AGE accumulation, adipogenesis is reduced, with the downregulated differentiation of fibroblasts and adipocytes [36]. Moreover, the accumulation of AGEs in the ECM also results in impaired glucose uptake and may contribute to the development of insulin resistance, as was shown in human primary adipocytes isolated from diabetic and non-diabetic obese individuals [37].

Furthermore, recent research highlights the potential role of the AGE–*RAGE* axis in regulating adaptive thermogenesis. This process activates temporary changes in white adipocytes in response to certain stressors, such as cold, exercise, or excessive energy input. Technically, it results in the stimulation of β3-adrenergic receptors, leading to the overexpression of uncoupling protein 1 (UCP1) and thereafter “beiging” of white adipocytes [38]. Activation of the *RAGE* gene in mice fed high-fat diets (HFDs) was found to downregulate adipose tissue browning and thermogenic activity compared with controls on a normal chow diet. Thus, the global deletion of *RAGE* in mice protects from this effect, even during exposure to an HFD [24,39]. Experiments on *RAGE−/−* murine adipocytes strongly supports this evidence, indicating that *RAGE* inhibits thermogenesis in WAT and BAT via diminishing the protein kinase A (PKA)-mediated phosphorylation of hormone-sensitive lipase (HSL) and other pathways (e.g., p38 mitogen-activated protein kinase, MAPK) involved in the regulation of energy balance [24].

### 3.2. Influence of the RAGE Pathway on Lipolysis/Lipogenesis

Molecules of major importance involved in adipose tissue metabolism are fatty acids (FAs). They are one of the two products of the hydrolytic degradation of triglycerides (TGs): lipolysis regulated by catecholamines, insulin, and natriuretic peptides [40,41]. It is well established that an activated β3-adrenergic receptor leads to the mobilization of cAMP and mediates the PKA-dependent phosphorylation of hormone-sensitive lipase (HSL) and p38 mitogen-activated protein kinase (MAPK), enzymes directly associated with lipolysis [42]. In a *RAGE*-rich environment, this pathway is inhibited; thus, lipid droplet accumulation increases, causing obesity and its complications. This concept is based on a mouse model, in which the transplantation of adipose tissue from mice with a global overexpression of *RAGE* to wild-type (WT) mice promoted obesity and insulin resistance [24].

On the other hand, detailed analysis of preadipocyte cultures revealed that the stimulation of *RAGE* induces an increase in adipocyte size, accelerates their differentiation, and enhances higher expression of *RAGE* ligands (e.g., AGE, HMGB1, S100β, and FA). In the preadipocytes transfected with small interfering RNA (siRNA), knockout of the *RAGE* gene led to the suppression of *RAGE* ligands and thus inhibited cell hypertrophy [13]. Morphological changes in adipocytes might also be mediated via crosstalk between Toll-like receptors (TLRs). This theory was followed by a series of studies on mice and cell cultures indicating that the downregulation of Toll-like receptor 2 (TLR2) and Toll-like receptor 4 (TLR4) in diet-induced obesity improves adipocyte differentiation and insulin sensitivity in mice [43,44]. However, experiments on mice fed an HFD and murine adipocytes revealed that the suppression of TLR2 enhances adipogenesis; TLR4 results in an opposite effect [44,45]. This dichotomy opens paths for future studies exploring *RAGE*–TLR linkages. In addition, the activation of *RAGE* promotes a switch in macrophages infiltrating the adipose tissue of mice with diet-induced obesity to M1-type producing inflammatory cytokines, such as tumor necrosis factor-alpha (TNF-α) and interleukin (IL) 1β [13,46]. Consequently, the route to the synthesis of ROS and the blockade of the phosphoinositide 3-kinase (PI3K)–protein kinase B (AKT) pathway promoting lipogenesis is open.

Another function of AGEs relevant to lipid metabolism is the inhibition of apolipoprotein E (ApoE) expression. ApoE, as a component of very low density lipoproteins (VLDLs), remnant lipoproteins, and high-density lipoproteins, contributes to lipoprotein internalization and degradation [47,48]. Using 3T3-L1 cells and mouse models, researchers observed that in hyperglycemic conditions, oxidative stress and the NF-κB pathway mediate ApoE abolishment. Blocking ApoE could partially participate in suppressing adipocyte triglyceride synthesis because the transplantation of wild-type adipocytes in mice with a global ApoE knockout results in less triglyceride accumulation. Additionally, ApoE knockdown adipocytes have fewer VLDL receptors on their cell surface, which results in the reduced internalization of lipoproteins [48].

### 3.3. Influence of the RAGE Pathway on Adipose Tissue Insulin Resistance

The AGE–*RAGE* axis in adipose tissue also appears to play a key role in the development of obesity-related insulin resistance. Potential pathogenetic mechanisms include, but are not limited to, altered insulin signaling, the promotion of adipocyte hypertrophy and related functional complications, the exacerbation of macrophage-mediated inflammation, impaired adipose tissue browning, and the dysregulation of adipokine secretion [49].

Animals with global knockout of the *RAGE* gene (*RAGE*−/−) and exposed to an HFD were characterized by improved glucose tolerance compared with wild-type (WT) controls reared under the same conditions [50]. This effect may be related to increased insulin-induced protein kinase B (AKT) phosphorylation, critical for glucose–insulin crosstalk, in the adipose tissue of the *RAGE*^−/−^ animals compared with the control mice. In addition, the *RAGE*^−/−^ mice showed reduced levels of free fatty acids and glycerol compared with the WT mice [50]. Through *RAGE*-mediated AMPK downregulation of the AKT signaling pathway, AGEs can also decrease insulin sensitivity in other tissues crucial for glucose metabolism, such as the skeletal muscle [51]. *RAGE* lacks intracellular kinase signaling activity; therefore, the effects of receptor stimulation depend on the proteins binding to its intracellular domain. One example of such proteins is Diaphanous 1 (DIAPH1), a member of the formin family of Rho GTPase binding proteins with identified roles in cytokinesis, actin polymerization, cytoskeleton remodeling, and immune cell trafficking. Activation of the AGE/*RAGE*/DIAPH1 axis was found to be positively associated with the expression of genes reflecting impaired glucose metabolism and markers of inflammation in the subcutaneous adipose tissue of obese individuals [51]. Moreover, AGE-stimulated murine adipocyte hypertrophy, which is accompanied by downregulation of insulin sensitivity genes (e.g., glucose transporter type 4 and adiponectin), may contribute to diminished glucose uptake and impaired insulin signaling [13].

AGE-induced inflammation also plays a key role in the development of insulin resistance. Consistently, global *RAGE* knockout correlates with diminished macrophage migration to adipose tissue and lower systemic IL-6 concentrations in mice on an HFD [50]. Perigonadal adipose tissue (PGAT) and bone marrow derived from *RAGE*−/− mice on an HFD exhibit reduced expression of inflammatory markers typical for M1 macrophages compared with controls. In addition, macrophages in the PGAT of *RAGE*−/− mice were shown to express lower levels of the CD11c marker which, together with polarization into the anti-inflammatory M2 phenotype, results in the upgraded regulation of insulin secretion [52,53]. The effect of AGEs on TLR signaling in the context of insulin sensitivity may also be relevant because mice with suppressed TLR2 presented disruption of glycometabolic control, higher levels of TGs, and inflammatory molecules [45]. However, other studies report contrasting results linking deficiency in TLR2 with improvements in glucose uptake parameters [13,54]. Another mechanism by which the AGE–*RAGE* pathway may regulate insulin sensitivity is through its effect on brown adipose tissue expansion; as described above, *RAGE* deficiency is a strong indicator of WAT browning [24,55].

Interestingly, *RAGE* impacts on glucose homeostasis were found to be sex-dependent. Female *RAGE*−/− mice on an HFD showed significantly improved glucose and insulin tolerance compared with males. This finding was accompanied by the increased polarization of M2 macrophages and the expression of genes involved in browning in adipose tissue, as well as elevated insulin-induced AKT phosphorylation [56]. Finally, the AGE–*RAGE* axis can modulate the insulin sensitivity of adipose tissue by regulating the adipokines it secretes (described in detail in the following section) [49,57].

Given the role of *RAGE* activation in the development of insulin resistance, it is conceivable that silencing the signals transmitted by this receptor may represent a therapeutic strategy to improve insulin sensitivity in patients with impaired glucose tolerance. The use of *RAGE* antagonists could potentially improve insulin receptor signaling, increase the expression of genes encoding glucose transporters, and reduce intracellular inflammation not only in adipocytes but also in hepatocytes and myocytes, etc.

### 3.4. Influence of the RAGE Pathway on Adipokine Secretion

Another feature of adipose tissue as a very dynamic organ is the secretion of bioactive peptides and proteins, including leptin, adiponectin, IL-6, TNF-α, and other cytokines, contributing to metabolic homeostasis. These molecules with both pro- and anti-inflammatory activity maintain insulin sensitivity and regulate cardiovascular and reproductive functions [57,58]. The obesity-related dysregulation of the AGE–*RAGE* pathway has an impact on adipokine secretion, although the exact roles in this crosstalk are still not clarified.

Leptin, mostly produced in WAT and released into the bloodstream in concentrations proportionate to fat mass, plays a crucial role in preserving energy balance as a part of the negative feedback loop. Through the blood–brain barrier (BBB), it conveys information to the central nervous system regarding energy expenditure; therefore, it is essential for maintaining metabolic homeostasis. In obesity, levels of leptin are higher and transport across the BBB is altered, while activated inflammatory pathways promote central leptin resistance. Similarly to IR in diabetes, impaired response to leptin seems to be selective. It has been shown that in mice on HFD leptin, due to activation of the renal sympathetic nervous system, the elevation of blood pressure was stimulated. However, this failed to induce anorexigenic effects [59,60]. In mice with obesity-associated diabetes, leptin downregulation leads to overexpression of *RAGE* in β-cells and therefore inhibits insulin infusion and mediates AGE-elicited pancreatic islet apoptosis [61]. It was suggested that AGE formation during prolonged hyperglycemia could cause β-cell damage through insufficient leptin action and subsequent *RAGE* induction [61]. These phenomena result in ROS generation and endoplasmic reticulum (ER) stress response. AGEs with leptin inhibitory properties include glycolaldehyde-modified BSA (which binds *RAGE*) and oxidized low-density lipoproteins which, in turn, interact with the CD36 receptor, as shown in 3T3-L1 adipocytes and mouse epididymal adipocytes [16]. An important role in the interaction between the AGE–*RAGE* pathway and leptin is played by the peroxisome proliferator-activated receptor-γ (PPAR-γ) [62]. PPAR-γ has been discovered to co-regulate leptin gene expression, as a heterozygotic deletion of PPAR-γ in HFD mice increased leptin levels [63]. Proper PPAR-γ expression in murine adipose tissue hampers AGE formation and acts as a downregulator of *RAGE*, thus contributing to the prevention of AGE–*RAGE* axis-mediated cardiovascular disorders [63]. Further studies should also consider the role of the AGE–*RAGE* pathway in the development of hypothalamus inflammation and central leptin resistance [64]. It is also worth pointing out that post-translational modifications play an important role in the regulation of leptin function. For example, the phosphorylation of serine 491 is essential for leptin’s effects on hypothalamic α2AMPK activity, neuropeptide expression, food intake, and body weight [65]. Leptin is a peptide; therefore, one would expect glycation to modify its functions. However, to date, there are no data available in the literature on the effect of glycation of the leptin molecule on its ability to regulate adipogenesis and adipose tissue function.

Leptin levels as well as the expression of adiponectin are significantly modified in obesity. In a recent preclinical study, an inverse correlation between serum adiponectin and atherosclerosis in *RAGE*^−/−^ mice was reported [66]. Still under investigation is whether the *RAGE* impacts on adiponectin secretion are direct or occur through the regulation of adipose tissue inflammation. One study revealed that the stimulation of *RAGE* by its agonist N(ε)-(carboxymethyl)lysine (CML) resulted in a significant increase in inflammatory molecules’ expression (such as plasminogen activator inhibitor (PAI)-1 and IL-6) with a simultaneous reduction in adiponectin secretion by human preadipocytes [57]. This observation was followed by another experiment. Likewise, in the case of leptin, ROS generation is recognized as a promoter of adiponectin downregulation. Indeed, Maeda et al. reported that epithelium-derived factor (PEDF), which is a multifunctional, antioxidant glycoprotein, inhibits the AGE–*RAGE*-induced suppression of adiponectin mRNA levels in human visceral adipocytes through the suppression of nicotinamide adenine dinucleotide phosphate (NADPH) oxidase [67].

Data regarding the effect of the activation of the AGE–*RAGE* pathway on the expression of other adipokines are scarce. However, it is known that *RAGE*-mediated signaling may play a key role in the pro-inflammatory effects of visfatin [68]. In turn, resistin production by macrophages/monocytes in humans correlates with serum s*RAGE* concentrations [69].

The impact of the AGE–*RAGE* pathway on adipose tissue is summarized in Table 1.

## 4. RAGE Pathway in Animal and Human Obesity and Related Complications

### 4.1. RAGE Pathway in Animal Models of Obesity

Two characteristics that distinguish obesity from other diseases are its worldwide prevalence and complexity, as well as the involvement of various organs, resulting in more than 200 complications [70]. Due to its high heterogeneity, novel therapies characterized by safe and personalized approaches are required in the treatment of obesity. Animal models, especially rodents, play an essential role in understanding the pathogenesis of obesity and finding new therapeutic solutions. Due to their susceptibility to nutritional interventions and the possibility of genetic modifications, they enable us to mimic different models of obesity and metabolic disorders. Nevertheless, to date, there is no ideal animal model to replicate human obesity. Factors that cause this imperfection include the different pathomechanisms of obesity in rodents compared with humans (e.g., different contributions of thermogenesis to energy balance) and the fact that experimental protocols are often based on diets with excessive fat contents or exposure to high temperatures, disrupting macro- and micronutrient balance and promoting bias in the methodology [71,72].

Given that the *RAGE* pathway has been proposed to participate in the development and progression of obesity, a number of preclinical studies exploring this hypothesis have emerged. As described before, *RAGE*^−/−^ mice on an HFD exhibit improved glucose tolerance and decreased lipolysis compared with WT animals, while females are also characterized by the reduced infiltration of adipose tissue by pro-inflammatory macrophages and the more intense browning of white adipocytes [24,50]. However, in other experimental settings, *RAGE*^−/−^ mice exposed to an HFD were characterized by the potential of developing clusters of metabolic syndrome (MetS) manifested by accelerated weight gain, increased plasma cholesterol, and higher insulin levels compared with control animals [73,74]. In addition, *RAGE*^−/−^ mice exhibited lower expression of the genes encoding antioxidative enzymes (Mn and Cu/Zn superoxide dismutases) and ceruloplasmin in cardiac tissue [74]. In contrast, in a study by Hoffman et al., the hearts and aortic valves of *RAGE*^−/−^ mice exposed to an HFD exhibited fewer morphometric changes, less calcification, and less AGE accumulation compared with WT C57BL/6 mice bred under the same conditions. Moreover, *RAGE*^−/−^ mice had a more favorable high-density to low-density lipoprotein (HDL/LDL) ratio, with inflammatory and oxidative stress parameters protecting them from ventricular remodeling [75,76]. These findings are consistent with data on the contribution of the AGE–*RAGE* pathway to the activation of inflammatory pathways, and the acceleration of dendritic cell maturation and cytokine production, which leads to the overexpression of genes causing cardiomyocyte hypertrophy and heart failure [77].

Animal models of obesity also indicate that the inflammatory state resulting from the activation of the AGE–*RAGE* pathway is implicated in the development of hyperglycemia-associated complications, such as retino-, neuro-, and nephropathy, as well as an increased risk of bone fracture and osteoporosis [78,79]. These observations suggest that blockade of the AGE–*RAGE* pathway may be a valuable therapeutic strategy for the future prevention and treatment of the chronic complications of diabetes.

### 4.2. The RAGE Pathway in Human Obesity

The growing awareness that *RAGE* plays a pivotal role in the development and progression of human obesity, as well as the knowledge regarding adipose tissue as an active endocrine organ, has led to the design of many experiments defining the functions of *RAGE*.

Attempts have been made to assess to what extent different lifestyle interventions might influence AGE–*RAGE* pathway activity. Using an alternate-day fasting (ADF) diet as a model of weight loss, changes in soluble *RAGE* isoforms (endogenous soluble *RAGE*, es*RAGE*, and cleaved *RAGE*, c*RAGE*) and adipokines were identified. This model assumed an intake of 25% of baseline caloric needs on fasting days and 125% of individuals’ energy needs on non-fasting days. As a result, es*RAGE* and the c*RAGE*:es*RAGE* ratio were significantly higher after 24 weeks of ADF in the experimental group. In addition, at this time point, changes in adipose tissue mass were negatively correlated with the es*RAGE* form. More interestingly, when considering fat deposits, a loss of subcutaneous adipose tissue was associated with a significant increase in es*RAGE* and the simultaneously constant level of c*RAGE*, suggesting es*RAGE* as an important factor in weight changes across fat deposits. As an additional effect, ADF resulted in moderate alterations in adipokine concentrations including negative correlations between IL-6, leptin, and s*RAGE* with no influence on total adiponectin level. However, changes in adiponectin concentration were inversely correlated with the c*RAGE*:es*RAGE* ratio [80]. Changes in the concentrations of soluble *RAGE* isoforms appear to be very dynamic; therefore, data from other studies revealed a positive correlation between es*RAGE* and adiponectin levels in obese female subjects [81]. Obesity-induced reductions in adiponectin and es*RAGE* levels have been conversely associated with markers of lipid peroxidation and platelet activation. Therefore, es*RAGE* may act as an antagonist of platelet activation, and its concentration may be helpful in assessing the cardiovascular risk of obesity [81]. In contrast, higher s*RAGE* levels are positively correlated with the mortality risk from cardiovascular diseases, whereas increased total *RAGE* concentrations indicate a greater possibility of death from all causes [82].

The concept of the involvement of the AGE–*RAGE* pathway in cardiovascular disease has inspired numerous studies on the association between epicardial adipose tissue (EAT) and s*RAGE* in this context. Studies have shown that in cardiometabolic diseases, the increasing EAT thickness and the increased expression of inflammatory genes are driven by the higher expression of *RAGE*; this is more pronounced in patients with diabetes. In addition, s*RAGE* as a decoy receptor is negatively associated with the EAT volume, waist circumference, and visceral adipose tissue deposits in obese but healthy women. Therefore, s*RAGE* may be a marker for cardiometabolic diseases, correlating with fat accumulation and visceral and epicardial deposits, associated with increased cardiovascular risk [83]. Subsequently, AGE and s*RAGE* concentrations and their potential as indicators of vasculopathy have been investigated in several studies across different age groups [84,85]. These studies show that the increased deposition of cholesterol and its derivatives or other markers of vascular disorders in arterial walls was associated with a decrease in s*RAGE* expression [86,87].

Considerable research has also been dedicated to the association of s*RAGE* and its forms with the risk of developing type 2 diabetes and its complications [12,78,88,89]. It was found that s*RAGE* and c*RAGE* were negatively correlated with plasma glucose levels. The decrease in c*RAGE* concentration progressed in patients with impaired glucose tolerance and diabetes mellitus, which was probably due to the hyperglycemia-induced enhanced proteolytic degradation of the *RAGE* subdomain mediated by disintegrins and metalloproteinases. The loss of es*RAGE* alone, on the other hand, was particularly marked in obese patients, in a manner proportional to BMI and percentage body fat. Age appeared to be a significant factor influencing s*RAGE* isoform concentrations. With age, differences in the concentrations of its isoforms disappear, but s*RAGE* and c*RAGE* concentrations were still reduced in people with type 2 diabetes [90]. In addition to attempts to understand the role of the AGE–*RAGE* axis in diabetes progression, the impact of this pathway on hyperglycemia-mediated calcification has been investigated. AGEs have been shown to mediate the activation of cellular and systemic responses leading to the activation of protein kinase C (PKC), p38 mitogen-activated protein kinase (MAPK), fetuin-A, and the NF-kB and ERK1/2 pathways upregulating extant bone matrix proteins, therefore promoting vascular calcification. AGE–*RAGE*-pathway-induced oxidative stress is also implicated in a phenotype shift of vascular smooth muscle cells to osteoblast-like cells [91].

Shedding light on the relationship between the AGE/*RAGE* axis and obesity, attention needs to be paid not only to endogenous forms but also dietary AGEs (dAGEs) and their impact on anthropometric measures. Significant negative associations between dAGE consumption and BMI, waist circumference, waist-to-hip ratio, fat-free mass, and muscle mass index in non-linear models have been described [92]. Similarly, a recent meta-analysis showed a negative correlation between circulating AGEs and BMI [93]. In turn, some researchers claim that diets with low intake of AGEs reduce BMI, with no such impact on waist circumference. This paradox might be explained using CML as a marker of AGEs, which, in obesity, is trapped in adipose tissue and subsequently causes falsely reduced levels of serum CML. Therefore, the suggested AGE type for further studies is methylglyoxal-derived hydroimidazolone-1 (MG H1) [94]. Low intake of AGEs is considered a preventative measure for obesity and its associated complications. Firstly, a recent meta-analysis revealed that diets with low levels of AGEs significantly decreased insulin resistance, fasting insulin, and total and LDL cholesterol, best reflected in subjects on prolonged low-AGE diets and with metabolic syndrome risk factors [95]. Secondly, low AGE consumption lowers leptin levels, with parallel increases in adiponectin concentration, which is consistent with previous studies highlighting the role of dAGEs in insulin resistance. Due to the heterogeneity of the diet, it is difficult to make precise recommendations on the optimal intake of dAGEs, but limiting the consumption of dAGEs may be a beneficial factor in supporting a healthy lifestyle [94,95]. In addition, the heterogeneity of the signals transmitted by *RAGE* means that the therapeutic effects associated with the blockade of the AGE–*RAGE* pathway can sometimes be difficult to predict.

Figure 2 summarizes in a simplified way how AGEs, by inducing adipose tissue dysfunction, may contribute to the development of obesity-related complications.

## 5. Therapeutic Perspective of Interfering with the RAGE Pathway to Counteract Obesity

A key feature to resolve the interplay between the *RAGE* pathway and the pathogenesis of obesity and related comorbidities is uncovering AGE receptor functions that could be exploited therapeutically. Dietary interventions remain the primary method for regulating AGE concentrations.

There is rapidly growing awareness of the impact of food nutritive value and daily calorie intake on health [96,97]. Thus, previous studies have demonstrated that reduced energy intake acts against AGE accumulation. Only approximately 10% of exogenous AGEs are absorbed from the diet; therefore, caloric restrictions have been found to be an important factor in suppressing *RAGE* mRNA levels, oxidative stress, and inflammatory responses in tissues, although the data are conflicting [98,99,100,101,102]. Accordingly, as described, weight loss might also have a beneficial impact on s*RAGE* concentration and adipocyte dysfunction. Importantly, the quantity as well as the quality of the diet contribute to AGE formation. Hence, Mediterranean diets based on monounsaturated fatty acids (MUFAs) and plant-based ingredients and prepared at lower temperatures and in higher humidities, resulting in lower doses of dAGEs, are preferred over Western diets rich in saturated fatty acids (SFAs) and highly processed, fried, or grilled food. In this setting, the daily intakes of PUFAs and MUFAs in the Mediterranean diet show antiglycation and antioxidant properties, such as decreased serum levels of AGEs and the expression of *RAGE* [101,102]. In addition, gene–diet interactions can modify metabolic parameters. For instance, via fatty acid desaturase 2 (FADS2) gene polymorphism, which significantly interacts with weight, adipose tissue mass, waist circumference, and cholesterol values and heterozygotes tend to exhibit lower accumulation of dAGEs [103].

Several studies have indicated that high daily concentrations of AGEs, especially in Western diets, correlate with the development and progression of multiple chronic diseases. Therefore, the effect of low dietary AGEs on biomarkers of these health disturbances was investigated. For example, there is evidence demonstrating that low intake of AGEs might decrease the level of 8-isoprostanes (markers of oxidative stress), particularly in patients with diabetes and chronic kidney disease [104,105]. Additionally, vascular cell adhesion molecule 1 (VCAM-1) and oxidized LDL concentrations, markers of cardiovascular diseases, are increased with high-AGE diets in hyperglycemic conditions, which might underpin a positive role of AGE-restricted diets. However, this effect was only observed with a small sample size, and further studies are necessary to support this consideration [106].

Actions aimed at lowering AGE concentrations do not solely result in beneficial cardiometabolic effects. AGEs also interact with periodontal tissues in diabetes and suppress the osteogenic differentiation ability of human periodontal ligament stem cells (hPDLSCs), leading to their degeneration. Wnt/β-catenin is considered to be the activated pathway; however, berberine hydrochloride, proposed as an inhibitor of this pathological process, may recover the differentiation abilities of hPDLSCs and can thus be considered an anti-AGE agent, useful in the treatment of diabetes-related periodontitis [107].

Given the clear negative correlation between *RAGE* isoforms and cardiometabolic indices, targeting their anti-inflammatory, antioxidative effects might be a step-up approach for the treatment of obesity and its comorbidities [108]. However, the current state of knowledge does not enable precise determination of the effect of long-term calorie restrictions and diets with different AGE contents on cardiometabolic risk and survival. Moreover, the results of randomized and observational studies are often contradictory. Conclusions from meta-analyses on therapeutic interventions interfering with the *RAGE* pathway to counteract obesity are summarized in Table 2.

## 6. Final Remarks and Conclusions

In recent years, knowledge of the *RAGE* pathway and its crosstalk with obesity and related complications has been the source of several studies. Thus far, it has been outlined that collaboration between AGEs and their receptors affects adipose tissue through modification of the development, functionality, and cleavage of adipose cells. The mechanisms that drive such effects still require investigation; however, preclinical research indicates a dominant role in inflammatory and oxidative stress pathways. This approach is consistent with concepts indicating obesity as a chronic low-grade inflammation disease [109].

An additional key finding is the influence of AGEs on other processes such as glucose–insulin homeostasis or adipokine secretion. For example, the AGE–*RAGE* axis can serve as a provoker of insulin resistance and altered leptin secretion, leading to an increased risk of diabetes and cardiovascular diseases’ prevalence. Thus, it will be challenging to identify the exact mechanism of action that could inhibit these pathological responses.

Finally, future studies should focus on expanding the therapeutical pipeline for obesity and related comorbidities with the use of the AGE–*RAGE* axis. Currently available therapeutic strategies targeting the AGE–*RAGE* pathway are based on limiting the supply of exogenous AGEs from the diet. An alternative option would be to reduce the synthesis of endogenous AGEs. In contrast, selective *RAGE* antagonists constitute one option that targets this pathway regardless of the AGE source.

## Figures and Tables

**Figure 1 ijms-24-10982-f001:**
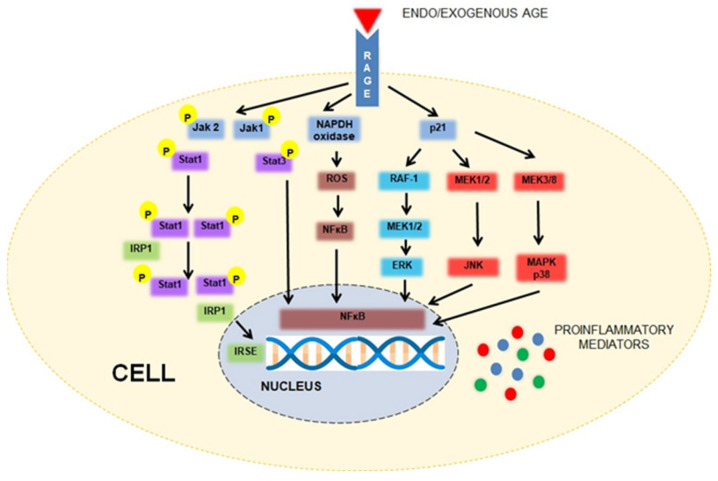
A simplified scheme of *RAGE*-mediated signaling. The interaction between the advanced glycation end products (AGEs) and their receptor (*RAGE*) stimulates several intracellular signaling cascades, e.g., Jak/Stat, NADPH oxidase, mitogen-activated protein kinases (MAPK)/p38, extracellular regulated (ERK)-1/2, and c-JunN-terminal kinase (JNK). These phenomena result in the activation of transcription factors, such as nuclear factor (NF-kB) or IFN-stimulated response elements (ISRE), enhancing the expression of proinflammatory mediators (modified, based on [8,23]).

**Figure 2 ijms-24-10982-f002:**
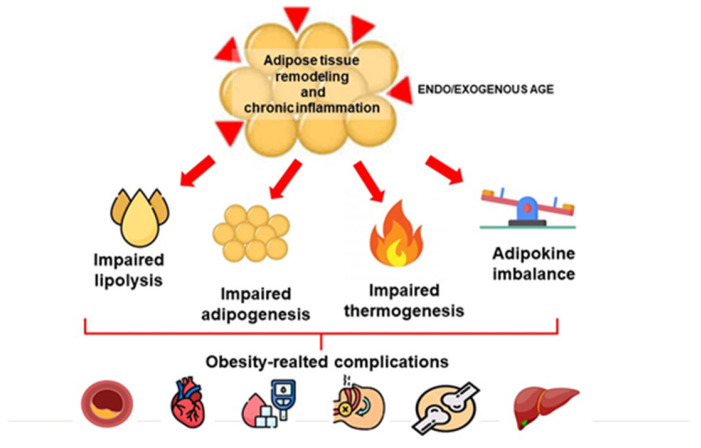
Effects of advanced glycation products (AGEs) on adipose tissue dysfunction and the development of obesity-related complications.

**Table 1 ijms-24-10982-t001:** The impact of advanced glycation end products (AGEs) on adipose tissue.

Process	Experimental Model	Effect of AGE–*RAGE* Pathway Activation	Mechanism	References
Adipogenesis	Human MSCs	↓ differentiation potential towards adipocytes		[10]
	ASCs from diabetic osteoporotic and control C57BL/6 mice	↓ proliferation ↓ differentiation potential of ASCs	↓ Wnt signaling pathway ↑ methyltransferase genes	[34]
	Senescent murine preadipocytes	↑ senescent preadipocytes differentiation	↓ p53 protein	[35]
Browning and thermogenesis	*RAGE*^−/−^ mice *RAGE*^−/−^ murine adipocytes	↓ thermogenesis	↓ PKA-mediated phosphorylation of HSL and p38 MAPK	[24]
	Mice on an HFD	↓ browning ↓ thermogenesis		[39]
Lipolysis	*RAGE*^−/−^ mice	↓ lipolysis ↑ weight gain	↓ PKA-mediated phosphorylation of HSL	[24]
Lipogenesis	DIO mice receiving a *RAGE* inhibitor	↑ lipogenesis	↑ TLR receptors ↓ PI3K protein kinase B pathway	[46] [48]
Insulin sensitivity	*RAGE*^−/−^ mice	↓ insulin sensitivity	↓ PI3K protein kinase B pathway ↑ DIAPH1 expression ↑ metabolic inflammation	[13,50] [51] [50]
Adipokine secretion	3T3-L1 adipocytes	↓ leptin secretion	↑ ROS synthesis ↓ PPAR-γ expression	[16] [63]
	*RAGE*^−/−^ mice	↓ adiponectin secretion	↑ ROS synthesis ↑ metabolic inflammation ↓ NADPH oxidase	[57] [67]

↓, decrease; ↑, increase; ASCs, adipose-derived stem cells; DIAPH1, diaphanous 1 protein; DIO, diet-induced obesity; HFD, high-fat diet; HSL, hormone-sensitive lipase; MAPK, mitogen-activated protein kinase; NADPH, nicotinamide adenine dinucleotide phosphate; PKA, protein kinase A; PI3K, phosphoinositide 3-kinase; PPAR-γ, peroxisome proliferator-activated receptor-γ; *RAGE*, receptor for advanced glycation end products; ROS, reactive oxygen species.

**Table 2 ijms-24-10982-t002:** Therapeutic interventions interfering with the *RAGE* pathway to counteract obesity.

Intervention	Participants	Duration	Outcome	References
Meta-Analyses				
Low-calorie diet 3 RCTs	123 overweight/obese patients with/without T2D	2–3 months	↓ AGE serum levels	[100]
Mediterranean diet rich in MUFAs and low in AGE 6 RCTs	395 overweight/obese patients with/without T2D	1–3 months	↓ AGE serum levels ↓ *RAGE* expression	[101]
Low-AGE diet 6 RCTs	172 overweight/obese patients with/without T2D	1 day to 12 weeks	↓ triglycerides ↓ fasting insulin and glucose ↓ uACR,↓ 8-isoprostanes ↓ BMI, WC, and WHR ↑ eGFR and HDLs	[102]
Low-AGE diet 13 RCTs	293 patients of different weights with/without T1D or T2D	2–16 weeks	↓ AGE, TNF-α, 8-isoprostanes, VCAM-1 and oxLDL serum levels	[106]

↓, decrease; ↑, increase; AGE, advanced glycation end products; BMI, body mass index; eGFR, estimated glomerular filtration rate; HDLs, high-density lipoproteins; MUFAs, monounsaturated fatty acids; oxLDLs, oxidated low-density lipoproteins; *RAGE*, receptor for advanced glycation end products; RCT, randomized controlled trial; T1D, type 1 diabetes mellitus; T2D, type 2 diabetes mellitus; TNF-α, tumor necrosis factor-α; uACR, urinary albumin/creatinine ratio; VCAM-1, vascular cell adhesion molecule 1; WC, waist circumference; WHR, waist-to-hip ratio.

## Data Availability

Not applicable.

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
