# Peer review of "Receptor for the Advanced Glycation End Products (*RAGE*) Pathway in Adipose Tissue Metabolism"

_ijms, 2023, doi:10.3390/ijms241310982_

Round 1

Reviewer 1 Report

The Manuscript (Review) entitled “Receptor for advanced glycation end-products (RAGE) pathway in adipose tissue metabolism” by Gutowska et al., have discussed the role of RAGE pathway om adipose tissue metabolism. This review article is well written in crisp manner to address the readers. However, this reviewer has certain query which is needed to be replied in point-by-point manner-

Comments

1.       Authors are advised to revise the whole manuscript in terms English language and grammar.

2.       There are also many typographical and spelling mistakes which should be thoroughly looked into.

3.       There are no figures in this review article. Figures and artwork help the readers, reviewers, editors to better understand the review literature. I would suggest the Authors to include atleast one figure in this review.

4.       Line 52-53: Kindly correct the sentence “as its disruption u results in adipocyte necrosis”.

5.       Line 82-84: kindly incorporate citation at the end of sentence “First, unstable Schiff bases are formed through direct correlation between carbonyl groups of reducing sugars and free amino groups of nucleic acids, proteins, or lipids [cite ref 1 & 2].

Ref. 1: Role of advanced glycation end products in cardiovascular disease. World journal of cardiology. 2012;4(4):90-102.

Ref. 2: Antiglycation study of HMG-R inhibitors and tocotrienol against glycated BSA and LDL: A comparative study. International journal of biological macromolecules. 2018;116:983-92.

6.       Line 133-136: kindly check spelling and grammar mistake here. This reviewer is not able to understand what the authors want to convey. This happens when authors try to re-phrase the sentence to avoid similarity with the previous work and end-up with something this unexplanatory sentences. Kindly avoid this.

7.       Leptin is a peptide hormone which has a role in adipose tissue metabolism. Since leptin is a peptide, does the glycation of leptin which have impact on Adipogenesis? Either it does or doesn’t Kindly mention this in your review.

English language specially grammar and sentence formation in not up-to the standard.

Author Response

Reviewer 1

We wanted to express our gratitude to the Reviewer for his constructive criticism and substantive support. We respond to his valid comments below.

Comments 1 & 2

  1. Authors are advised to revise the whole manuscript in terms English language and grammar.
  2. There are also many typographical and spelling mistakes which should be thoroughly looked into.

We apologise for the careless editing of our article. As suggested by the reviewer, the manuscript has been sent to a professional English-language editor (please see attached certificate).

Comment 3

  1. There are no figures in this review article. Figures and artwork help the readers, reviewers, editors to better understand the review literature. I would suggest the Authors to include at least one figure in this review.

Following the Reviewer's valuable comment, we have added two figures to the manuscript: Figure 1 on the molecular pathways involved in the AGE receptor and Figure 2 summarising the potential effects of overexpression and activation of this receptor in adipose tissue.

Figure 1. A simplified scheme of RAGE-mediated signaling. The interaction between the advanced glycation end products (AGEs) and their receptor (RAGE) stimulates several intracellular signaling cascades, e.g. Jak/Stat, NADPH oxidase, mitogen activated protein kinases (MAPK)/p38, extracellular regulated(ERK)-1/2 and c-JunN-terminal kinase (JNK). These phenomena result in the activation of transcription factors, such as nuclear factor (NF-kB) or IFN-stimulated response elements (ISRE) enhancing expression of proinflammatory mediators (modified, based on 8,23). lines 162-167.

Figure 2: Effects of advanced glycation products (AGEs) on adipose tissue dysfunction and the development of obesity-related complications. Lines 513-514

Comment 4

  1. Line 52-53: Kindly correct the sentence “as its disruption u results in adipocyte necrosis”.

We thank the reviewer for drawing attention to this typo, which has been removed in the revised version of the manuscript.

An important component of normal adipose tissue expansion is the formation of new blood vessels, as this disruption results in adipocyte necrosis, as well as hypoxia, which trigger chronic low-grade inflammation and fibrosis, exacerbating adipose tissue dysfunction and leading to increased insulin resistance [6]. Lines 64-68.

Comment 5

  1. 5.Line 82-84: kindly incorporate citation at the end of sentence “First, unstable Schiff bases are formed through direct correlation between carbonyl groups of reducing sugars and free amino groups of nucleic acids, proteins, or lipids [cite ref 1 & 2].

The reviewer's suggested references have been added to the relevant paragraph in the manuscript [References 18 & 19].

First, unstable Schiff bases are formed through direct correlation between carbonyl groups of reducing sugars and free amino groups of nucleic acids, proteins, or lipids. If amino acids are included, this can lead to disturbances in the structure and function of proteins crucial for cells’ organization and metabolism [19-21]. Lines 115-119

[20] Hegab, Z.; Gibbons, S.; Neyses, L.; Mamas, M. A. Role of advanced glycation end products in cardiovascular disease. World J Cardiol 2012, 4, 90–102. doi:10.4330/wjc.v4.i4.90

[21] Nabi, R.; Alvi, S. S.; Khan, R. H.; Ahmad, S.; Ahmad, S.; Khan, M. S. Antiglycation study of HMG-R inhibitors and tocotrienol against glycated BSA and LDL: A comparative study. Int J Biol Macromol 2018, 116, 983–992. https://doi.org/10.1016/j.ijbiomac.2018.05.115

Comment 6

  1. 6.Line 133-136: kindly check spelling and grammar mistake here. This reviewer is not able to understand what the authors want to convey. This happens when authors try to re-phrase the sentence to avoid similarity with the previous work and end-up with something this unexplanatory sentences. Kindly avoid this.

We thank the Reviewer for this important comment. We have rephrased the indicated paragraph (and other parts of the text) to be clearer.

Several studies have supported the hypothesis that, in addition to the full-length form at the cell surface, RAGE also exist in other non-membrane-bound forms. Soluble RAGE (sRAGE) are produced by either the proteolytic cleavage of the full-length form of RAGE (cRAGE) or alternative mRNA splicing, and lack trans-membrane and cytoplasmic domains. Importantly, sRAGE can inhibit full-length RAGE activation by binding to their ligands in the extracellular space [29]. Lines 168-173.

Comment 7

  1. Leptin is a peptide hormone which has a role in adipose tissue metabolism. Since leptin is a peptide, does the glycation of leptin which have impact on Adipogenesis? Either it does or doesn’t Kindly mention this in your review.

We thank the reviewer for this important comment. Post-translational modifications can indeed affect leptin function. For example, phosphorylation of serine 491 is essential for leptin's effects on hypothalamic α2AMPK activity, neuropeptide expression, food intake, and body weight [PMID: 22727014]. However, we found no data in the available literature on the effect of glycation of the leptin molecule on its ability to regulate adipogenesis and adipose tissue function. This information was included in the revised version of the manuscript.

It is also worth pointing out that the post-translational modifications play an important role in the regulation of leptin function. For example, the phosphorylation of serine 491 is essential for leptin's effects on hypothalamic α2AMPK activity, neuropeptide expression, food intake, and body weight [68]. Leptin is a peptide; therefore, one would expect glycation to modify its functions. However, to date, there are no data available in the literature on the effect of glycation of the leptin molecule on its ability to regulate adipogenesis and adipose tissue function.Lines 357-363

[68] Dagon, Y.; Hur, E.; Zheng, B.; Wellenstein, K.; Cantley, L. C.; Kahn, B. B. p70S6 kinase phosphorylates AMPK on serine 491 to mediate leptin's effect on food intake. Cell Metab 2012, 16, 104–112. https://doi.org/10.1016/j.cmet.2012.05.010

Reviewer 2 Report

The manuscript is well written and comprises an interesting review topic, divided in such a way to ease full understanding of what's being presented.

A few observations are shown below, divided by topics

Affiliations

There’s a 4th affiliation, but it is not linked to any of the authors.

Affiliation 3 seems to be in a larger font than the others.

Keywords

Maybe adding both advanced glycation end-products (AGEs) and receptor for advanced glycation end-products (RAGE) as keywords could be redundant. One of them could be replaced for another word not present in the manuscrit's title.

Abstract

Advanced glycation end products writing in the beginning of the abstract is different from advanced glycation end-products in the title.

Introduction

I believe there is a typo @ the end of line 52 (there’s a letter u by itself)

The term @ line 56 is written as Advanced glycation end products, different from the title. I suggest maintaining uniformity in its writing pattern throughout the manuscript.

Hyperglycemia (line 56) can be also found written as hyperglycaemia (line 61). I suggest maintaining uniformity in its writing pattern throughout the manuscript.

Topics 2 and 2.2 are named identically.

Topic 3

@ line 205, there’s a typo “swith”

Topic 5

Since this is a topic of sound relevance, i would attempt to produce a Table summarizing the therapeutic findings of the manuscript regarding interfering with RAGE pathway to counteract obesity.

Author Contributions

The formatting is not the same throughout this topic. Example: “Conceptualization:” vs “writing—original draft preparation,”

Funding

I believe spaces are required @ line 502: theNational ScienceCentre 

The manuscript is written with a good english quality. Only minor editing is necessary in order to maintain cohesion (discussed in the comments section).

Author Response

Reviewer 2:

The manuscript is well written and comprises an interesting review topic, divided in such a way to ease full understanding of what's being presented. A few observations are shown below, divided by topics

We would like to thank the reviewer for the positive reception of our work and the comments, which will undoubtedly improve the quality of the manuscript.

Comments 1 & 2

Affiliations

There’s a 4th affiliation, but it is not linked to any of the authors.

Affiliation 3 seems to be in a larger font than the others.

We are very sorry for the editing shortcomings of the manuscript. Some of these arose from our errors, and some (such as differences in font size in some paragraphs) arose from the formatting of the manuscript during the editing process. In addition, after revision, the manuscript has been sent to a professional English-language editor (please see attached certificate).

Comment 3

Keywords

Maybe adding both advanced glycation end-products (AGEs) and receptor for advanced glycation end-products (RAGE) as keywords could be redundant. One of them could be replaced for another word not present in the manuscript's title.

In line with the Reviewer's suggestion, we have modified the Keywords section. To avoid redundancy, the term 'receptor for advanced glycation end products (RAGE)' has been replaced by 'AGE receptor (RAGE)' and 'adipose tissue' by 'adipose tissue dysfunction'. We have also added the terms: ‘insulin resistance’ and ‘cardiovascular diseases’.

Keywords: advanced glycation end products (AGEs); receptor for AGE (RAGE); obesity; adipose tissue dysfunction; metabolic inflammation; insulin resistance; diabetes; cardiovascular diseases. Lines 35-37.

Comment 4

Abstract

Advanced glycation end products writing in the beginning of the abstract is different from advanced glycation end-products in the title.

Once again, we must apologise for the careless editing of the manuscript. In its revised version, the nomenclature has been unified.

Comment 5

Introduction

I believe there is a typo @ the end of line 52 (there’s a letter u by itself)

We thank the reviewer for drawing attention to this typo, which has been removed in the revised version of the manuscript.

An important component of normal adipose tissue expansion is the formation of new blood vessels, as this disruption results in adipocyte necrosis, as well as hypoxia, which trigger chronic low-grade inflammation and fibrosis, exacerbating adipose tissue dysfunction and leading to increased insulin resistance [6]. Lines 64-68.

Comment 6

The term @ line 56 is written as Advanced glycation end products, different from the title. I suggest maintaining uniformity in its writing pattern throughout the manuscript.

Once again, we must apologise for the careless editing of the manuscript. In its revised version, the nomenclature has been unified.

Comment 7

Hyperglycemia (line 56) can be also found written as hyperglycaemia (line 61). I suggest maintaining uniformity in its writing pattern throughout the manuscript.

In the revised version of the manuscript , the spelling has been unified (page 1, line 17; page 2, line 72; page 11, lines 426 and 433).

Comment 8

Topics 2 and 2.2 are named identically.

Following the Reviewer's suggestion, the title of Section 2 has been modified so that it is not identical to the title of Subsection 2.2.

  1. Advanced glycation end products (AGEs) and the receptor for advanced glycation end products (RAGE) pathway. Lines 92-93.

Comment 9

Topic 3

@ line 205, there’s a typo “swith”

We thank the reviewer for drawing attention to this typo, which has been removed in the revised version of the manuscript.

This dichotomy opens paths for future studies exploring RAGE–TLR linkages. In addition, the activation of RAGE promotes a switch in macrophages infiltrating the adipose tissue of mice with diet-induced obesity to M1-type producing inflammatory cytokines, such as tumor necrosis factor-alpha (TNF-α) and interleukin (IL) 1β [13,46]. Lines 255-259.

Comment 10

Topic 5

Since this is a topic of sound relevance, i would attempt to produce a Table summarizing the therapeutic findings of the manuscript regarding interfering with RAGE pathway to counteract obesity.

As suggested by the Reviewer, an additional table (Table 2) entitled “Therapeutic interventions associated with interfering with RAGE pathway to counteract obesity” has been prepared.

Conclusions from meta-analyses on therapeutic interventions interfering with the RAGE pathway to counteract obesity are summarized in Table 2. Lines 563-564.

Table 2. Therapeutic interventions interfering with the RAGE pathway to counteract obesity.

Intervention

Participants

Duration

Outcome

References

Meta-analyses

Low-calorie diet

3 RCTs

123 overweight/obese patients with/without T2D

2-3 months

↓ AGE serum levels

[103]

Mediterranean diet rich in MUFAs and low in AGE

 6 RCTs

395 overweight/obese patients with/without T2D

1-3 months

↓ AGE serum levels

↓ RAGE expression

[104]

Low-AGE diet

6 RCTs

172 overweight/ obese patients with/without T2D

1 day to 12 weeks

↓ triglycerides

↓ fasting insulin & glucose

↓ uACR ,↓8-isoprostanes

↓ BMI, WC, WHR

↑eGFR and HDLs

[105]

Low-AGE diet

13 RCTs

293 patients of different weights with/without T1D or T2D

2-16 weeks

↓ AGE, TNF-α, 8-isoprostanes, VCAM-1 and oxLDL serum levels

[109]

↓, decrease; ↑, increase; AGE, advanced glycation end products; BMI, body mass index; eGFR, estimated glomerular filtration rate; HDLs, high-density lipoproteins; MUFAs, monounsaturated fatty acids; oxLDLs, oxidated low-density lipoproteins; RAGE, receptor for advanced glycation end products; RCT, randomized controlled trial; T1D, type 1 diabetes mellitus; T2D, type 2 diabetes mellitus; TNF-α, tumor necrosis factor-α; uACR, urinary albumin/creatinine ratio; VCAM-1, vascular cell adhesion molecule 1; WC, waist circumference; WHR, waist-to-hip ratio.

Comments 11 & 12

Author Contribution. The formatting is not the same throughout this topic. Example: “Conceptualization:” vs “writing—original draft preparation,”

Funding    I believe spaces are required @ line 502: theNational ScienceCentre 

Once again, we must apologise for the inattentive editing of the manuscript. In the revised version, the formatting in the Author Contribution section has been standardised and the necessary spaces in the Funding section have been introduced.

Author Contributions: conceptualization: A.K.; writing—original draft preparation: K.G. and A.K.; writing—review and editing: A.K. and K.C.; funding acquisition: A.K. All authors have read and agreed to the published version of the manuscript. Lines 595-597.

Funding: This study was supported by the National Science Centre Poland, grant 2018/31/B/NZ5/01556.

Lines 599-600.

Reviewer 3 Report

This manuscript offers meaningful insights into the complex relationship between the RAGE pathway and obesity, illustrating its significance within the broader context of metabolic health and disease. Overall, the review manuscript is well organized and clearly presented. However, to further enhance the manuscript's coherence, depth, and impact, I have the following comments/questions listed by the sections:

Major:

1. section 2

·         Section 2.1, AGEs can form through multiple pathways. It could be beneficial to discuss how different physiological or pathological conditions might favor one pathway over another. For instance, are there circumstances under which the Maillard pathway is more prominent versus the Wolff or Namiki pathway?

·         Section 2.1, The manuscript delves into the classification of AGEs based on their origin and structure. However, it could be interesting to elaborate on whether these different classes of AGEs have distinct biological effects. Does the source of AGEs (endogenous vs. exogenous) or their structure (cross-linkers, non-fluorescent cross-linker products, or non-fluorescent, non-crosslinkers) impact their function or their ability to bind to RAGE?

·         Section 2.2, It's mentioned that RAGE can bind to multiple ligands, including AGEs and DAMPs. Are there differences in how these ligands interact with RAGE and activate intracellular signaling pathways? Do these different ligands have distinct biological effects?

2. Section 3

·         Section 3.3 has comprehensive information about the role of RAGE pathway in insulin resistance. The connection between AGE-RAGE and insulin sensitivity is quite interesting. It might be beneficial to include a more detailed discussion on the potential clinical implications of these findings, such as how modulation of RAGE pathway could be beneficial in treating insulin resistance.

·         section 3,4 focuses on how RAGE impacts the secretion of leptin and adiponectin. To provide a more comprehensive view, the authors may consider expanding this section to include the effect of the RAGE pathway on other adipokines, such as resistin or visfatin.

4. section 4

·         section 4.1, the RAGE pathway is implicated in hyperglycemia-associated complications such as retinopathy, neuropathy, and nephropathy, increasing the risk of bone fracture and osteoporosis. This suggests that the RAGE pathway could be a target for therapies aimed at managing these complications.

·         section 4.2, There are indications that the RAGE pathway plays a critical role in human obesity. Changes in RAGE isoforms due to lifestyle interventions (like alternate day fasting) show promise for treating obesity, and have potential effects on adipokines, which are involved in metabolic and inflammatory processes. However, the dynamics of RAGE isoforms are complex and require further exploration.

6. section 5

·         This section suggests that caloric restriction could counteract the accumulation of advanced glycation end products (AGEs) and suppress RAGE expression, oxidative stress, and inflammation. While this presents an interesting hypothesis, more research is required to determine the exact impact and viability of long-term caloric restriction, and to what extent these impacts are mediated by changes in the RAGE pathway.

·         This section presents a correlation between high dietary AGE intake, common in Western diets, and the development and progression of chronic diseases. Further discussion and references would be helpful to understand this correlation in more depth and its potential implications for treatment strategies.

6. section 6

·         The conclusion succinctly summarizes the main points. However, discussing potential future research directions and the implications of this research could enhance its impact.

7. More explicit identification of gaps in current research would be useful for directing future studies. Though the authors touch on several areas of uncertainty, a more structured approach to pointing out these research gaps would be beneficial.

I believe your manuscript provides valuable insights into the role of the RAGE pathway in obesity and related complications, and the potential for therapeutic interventions. These suggestions aim to enhance clarity, provide more comprehensive coverage of recent research, and improve the reader's understanding. I hope you find these comments helpful in revising your manuscript.

There are many simple typos throughout the manuscript. The authors should carefully go over the manuscript for proofreading.

Author Response

Reviewer 3

This manuscript offers meaningful insights into the complex relationship between the RAGE pathway and obesity, illustrating its significance within the broader context of metabolic health and disease. Overall, the review manuscript is well organized and clearly presented. However, to further enhance the manuscript's coherence, depth, and impact, I have the following comments/questions listed by the sections:

Section 2

Comment 1

Section 2.1, AGEs can form through multiple pathways. It could be beneficial to discuss how different physiological or pathological conditions might favor one pathway over another. For instance, are there circumstances under which the Maillard pathway is more prominent versus the Wolff or Namiki pathway?

In response to a valuable comment from a Reviewer, the paragraph on the different pathways of AGE synthesis has been supplemented to include situations when particular pathways are activated.

AGE formation depends on the concentration and reactivity of glucose, the amount of AGE precursors, and the availability of free amino groups. The Wolff (polyol) pathway is active under hyperglycemic conditions and involves the formation of sorbitol from glucose. Excessive activation of this pathway leads to increased dicarbonyl formation through the accumulation of upstream metabolites such as fructose. Endogenous lipid peroxidation, resulting from the ROS-induced lipid peroxidation of polyunsaturated fatty acids present in cell membranes, can also lead to the increased formation of dicarbonyls, and subsequently, AGEs (Namiki pathway). AGEs formed by reactive dicarbonyls produced by lipid peroxidation are also referred to as ALEs (advanced lipoxidation end products) [23]. Lines 129-136.

Comments 2 & 3

Section 2.1, The manuscript delves into the classification of AGEs based on their origin and structure. However, it could be interesting to elaborate on whether these different classes of AGEs have distinct biological effects. Does the source of AGEs (endogenous vs. exogenous) or their structure (cross-linkers, non-fluorescent cross-linker products, or non-fluorescent, non-crosslinkers) impact their function or their ability to bind to RAGE?

Section 2.2, It's mentioned that RAGE can bind to multiple ligands, including AGEs and DAMPs. Are there differences in how these ligands interact with RAGE and activate intracellular signaling pathways? Do these different ligands have distinct biological effects?

We thank the reviewer for these valuable comments. Indeed, according to the literature, the pathways activated upon RAGE binding are often cell and ligand-specific. We have added this information in the revised version of the manuscript.

The pathways activated upon RAGE binding are often cell- and ligand-specific. For instance, in murine macrophages (RAW cells), the stimulation of RAGE by glycated bovine serum albumin (BSA) leads to the activation of the Jak–Stat pathway, while the binding of RAGE with CML in rat mesangial cells results in the activation of angiotensin II signaling [23].A simplified scheme of RAGE-mediated signaling is shown in Figure 1. Lines 156-161.

Section 3

Comment 1·         

Section 3.3 has comprehensive information about the role of RAGE pathway in insulin resistance. The connection between AGE-RAGE and insulin sensitivity is quite interesting. It might be beneficial to include a more detailed discussion on the potential clinical implications of these findings, such as how modulation of RAGE pathway could be beneficial in treating insulin resistance.

We agree with the Reviewer that such an issue needs to be addressed more extensively, so we have added the following paragraph to the revised version of the manuscript.

Given the role of RAGE activation in the development of insulin resistance, it is conceivable that silencing the signals transmitted by this receptor may represent a therapeutic strategy to improve insulin sensitivity in patients with impaired glucose tolerance. The use of RAGE antagonists could potentially improve insulin receptor signaling, increase the expression of genes encoding glucose transporters, reduce intracellular inflammation not only in adipocytes, but also in hepatocytes and myocytes, etc. Lines 318-323.

Comment 2    

Section 3,4 focuses on how RAGE impacts the secretion of leptin and adiponectin. To provide a more comprehensive view, the authors may consider expanding this section to include the effect of the RAGE pathway on other adipokines, such as resistin or visfatin.

We thank the reviewer for this important comment. The data available in the literature on the regulation of the secretion of other adipokines by the AGE-RAGE pathway are scarce; nevertheless, we have added a relevant paragraph to the revised text.

Data regarding activation of the AGE–RAGE pathway on the expression of other adipokines are scarce. However, it is known that RAGE-mediated signaling may play a key role in the pro-inflammatory effects of visfatin [71]. In turn, resistin production by macrophages/monocytes in humans correlates with serum sRAGE concentrations [72]. Lines 378-381.

Section 4

Comment 1·       

Section 4.1, the RAGE pathway is implicated in hyperglycemia-associated complications such as retinopathy, neuropathy, and nephropathy, increasing the risk of bone fracture and osteoporosis. This suggests that the RAGE pathway could be a target for therapies aimed at managing these complications.

We fully agree with the reviewer's opinion and have therefore included a relevant comment in the revised version of the manuscript.

Animal models of obesity also indicate that the inflammatory state resulting from the activation of AGE–RAGE pathway is implicated in the development of hyperglycemia-associated complications, such as retino-, neuro-, and nephropathy, as well as increased risks of bone fracture and osteoporosis [81-82]. These observations suggest that blockade of the AGE–RAGE pathway may be a valuable therapeutic strategy for the future prevention and treatment of chronic complications of diabetes. Lines423-428.

Comment 2 

Section 4.2, There are indications that the RAGE pathway plays a critical role in human obesity. Changes in RAGE isoforms due to lifestyle interventions (like alternate day fasting) show promise for treating obesity, and have potential effects on adipokines, which are involved in metabolic and inflammatory processes. However, the dynamics of RAGE isoforms are complex and require further exploration.

Indeed, the heterogeneity of the signals transduced by RAGE, depending on which ligand it binds to and in which cell it is expressed, makes the therapeutic effects associated with blocking this pathway sometimes difficult to predict. For this reason, we have added the following commentary:

In addition, the heterogeneity of the signals transmitted by RAGE means that the therapeutic effects associated with blockade of the AGE–RAGE pathway can sometimes be difficult to predict. Lines 508-510.

Section 5

Comments 1 & 2   

This section suggests that caloric restriction could counteract the accumulation of advanced glycation end products (AGEs) and suppress RAGE expression, oxidative stress, and inflammation. While this presents an interesting hypothesis, more research is required to determine the exact impact and viability of long-term caloric restriction, and to what extent these impacts are mediated by changes in the RAGE pathway.

This section presents a correlation between high dietary AGE intake, common in Western diets, and the development and progression of chronic diseases. Further discussion and references would be helpful to understand this correlation in more depth and its potential implications for treatment strategies.

We fully agree with the reviewer that the current state of knowledge does not allow a precise determination of the effect of long-term calorie restriction and diets with different AGEs on cardiometabolic risk and survival. The results of randomised and observational studies are often contradictory. Meta-analyses are a reliable solution in this case - therefore, to facilitate the perception of this section, we have summarized their findings in a separate table (Table 2). 

However, the current state of knowledge does not enable precise determination of the effect of long-term calorie restrictions and diets with different AGE contents on cardiometabolic risk and survival. Moreover, the results of randomized and observational studies are often contradictory. Conclusions from meta-analyses on therapeutic interventions interfering with the RAGE pathway to counteract obesity are summarized in Table 2. Lines…559-564

Table 2. Therapeutic interventions interfering with the RAGE pathway to counteract obesity.

Intervention

Participants

Duration

Outcome

References

Meta-analyses

Low-calorie diet

3 RCTs

123 overweight/obese patients with/without T2D

2-3 months

↓ AGE serum levels

[103]

Mediterranean diet rich in MUFAs and low in AGE

 6 RCTs

395 overweight/obese patients with/without T2D

1-3 months

↓ AGE serum levels

↓ RAGE expression

[104]

Low-AGE diet

6 RCTs

172 overweight/ obese patients with/without T2D

1 day to 12 weeks

↓ triglycerides

↓ fasting insulin & glucose

↓ uACR ,↓8-isoprostanes

↓ BMI, WC, WHR

↑eGFR and HDLs

[105]

Low-AGE diet

13 RCTs

293 patients of different weights with/without T1D or T2D

2-16 weeks

↓ AGE, TNF-α, 8-isoprostanes, VCAM-1 and oxLDL serum levels

[109]

↓, decrease; ↑, increase; AGE, advanced glycation end products; BMI, body mass index; eGFR, estimated glomerular filtration rate; HDLs, high-density lipoproteins; MUFAs, monounsaturated fatty acids; oxLDLs, oxidated low-density lipoproteins; RAGE, receptor for advanced glycation end products; RCT, randomized controlled trial; T1D, type 1 diabetes mellitus; T2D, type 2 diabetes mellitus; TNF-α, tumor necrosis factor-α; uACR, urinary albumin/creatinine ratio; VCAM-1, vascular cell adhesion molecule 1; WC, waist circumference; WHR, waist-to-hip ratio.

Section 6

Comment 1·

The conclusion succinctly summarizes the main points. However, discussing potential future research directions and the implications of this research could enhance its impact.More explicit identification of gaps in current research would be useful for directing future studies. Though the authors touch on several areas of uncertainty, a more structured approach to pointing out these research gaps would be beneficial.

Following the reviewer's suggestion, we have added the following paragraph to the Conclusions section.

Currently available therapeutic strategies targeting the AGE–RAGE pathway are based on limiting the supply of exogenous AGEs from the diet. An alternative option would be to reduce the synthesis of endogenous AGEs. In contrast, selective RAGE antagonists constitute one option that targets this pathway regardless of the AGE source. Lines 589-593.

Comment 2·       

I believe your manuscript provides valuable insights into the role of the RAGE pathway in obesity and related complications and the potential for therapeutic interventions. These suggestions aim to enhance clarity, provide more comprehensive coverage of recent research, and improve the reader's understanding. I hope you find these comments helpful in revising your manuscript.

We would like to thank the reviewer for the positive reception of our work and the comments, which will undoubtedly improve the quality of the manuscript

  1. Comments on the Quality of English Language

There are many simple typos throughout the manuscript. The authors should carefully go over the manuscript for proofreading.

We apologise for the careless editing of our article. As suggested by the reviewer, the manuscript has been sent to a professional English-language editor (please see attached certificate).

Reviewer 4 Report

The manuscript by Gutowska et al. provides a summary of recent literature on AGE-RAGE signaling in adipose tissue. Overall, the review provides an important overview of this complex area of biology, however, the language needs to be tightened up and there are also several typos. I think the review needs revision, and a few suggestions are offered that might improve the review below. In general, the review is long and are trying to cover everything across all cell types in the adipose tissue. It would be beneficial for the reader if it was more simplistic/conceptualizing, and I think the authors should try to come up with a few key concepts (maybe only 1-3 cell types) discuss and graphically illustrate these ideas. 

1)    The review would benefit tremendously from having some graphical illustrations of the key concepts. I will leave it up to the authors what they think is most important to graphically illustrate, but for example, complex signaling cascades and biochemical pathways would be great to have as an illustration.

2)    I don’t think the authors can state: “… evidence for an AGE-RAGE pathway involvement in the regulation of adipogenesis” based on reference 31 as this paper only test the osteogenic differentiation. Also, this paper might suggest a positive effect on adipogenesis by AGEs in murine 3T3-L1 preadipocytes (PMID: 23150674), while this paper in human MSCs suggests a negative effect (PMID: 31875018).

3)    Similarly, references 33 or 34 do not measure adipocyte differentiation, but only glucose uptake. Consequently, this statement needs revision: “Recent study suggests that accumulation of AGEs in the ECM, independently of RAGE pathway, impaires adipocytes’ differentiation and metabolism [33,34]”. In fact, I think reference 34 might be wrong here.

4)    The authors should carefully state when conclusions/experiments/references are based on murine, human, or a combination.

For example, “Recent study suggests that accumulation of AGEs in the ECM, independently of RAGE pathway, impaires adipocytes’ differentiation and metabolism [33,34]”. The conclusion here is based on murine and human adipocytes in reference 33, and the sentence could be “Recent studies suggest that accumulation of AGEs in the ECM, independently of the RAGE pathway, impairs glucose uptake in mouse and human adipocyte models”.

5)    The adipose tissue is extremely heterogeneous. The authors should elaborate on which specific cells they are discussing, and when it is whole adipose tissue (a mix of all cell types). This also applies to genetic mouse models, when is it constitutive whole-body deletion, when is it constitutive cell-type specific and when is it inducible. This would be important information for the reader to interpret the summary.

6)    Further, I’m missing general information about which cells in the adipose tissue have RAGE receptors. This information should be possible to find in publicly available single cell/nuclei datasets. Further, is it clear which cells in the adipose tissue secrete AGE ligands, or are most AGEs exogenous?

The language needs to be tightened up and there are also several typos.

Author Response

Reviewer 4

The manuscript by Gutowska et al. provides a summary of recent literature on AGE-RAGE signaling in adipose tissue. Overall, the review provides an important overview of this complex area of biology, however, the language needs to be tightened up and there are also several typos. I think the review needs revision, and a few suggestions are offered that might improve the review below. In general, the review is long and are trying to cover everything across all cell types in the adipose tissue. It would be beneficial for the reader if it was more simplistic/conceptualizing, and I think the authors should try to come up with a few key concepts (maybe only 1-3 cell types) discuss and graphically illustrate these ideas. 

We wanted to express our gratitude to the Reviewer for his constructive criticism and substantive support. The heterogeneous cellular composition of adipose tissue makes it difficult to comprehensively describe the phenomena that occur in it under the influence of AGEs. Most of the published data concern adipocytes and therefore, in our work, we have focused primarily on describing studies on the effects of AGEs on adipocyte development and function. We respond to other valid comments below.

Comment 1

  • The review would benefit tremendously from having some graphical illustrations of the key concepts. I will leave it up to the authors what they think is most important to graphically illustrate, but for example, complex signaling cascades and biochemical pathways would be great to have as an illustration.

Following the Reviewer's valuable comment, we have added two figures to the manuscript: Figure 1 on the molecular pathways involved in the AGE receptor and Figure 2 summarising the potential effects of overexpression and activation of this receptor in adipose tissue.

Figure 1. A simplified scheme of RAGE-mediated signaling. The interaction between the advanced glycation end products (AGEs) and their receptor (RAGE) stimulates several intracellular signaling cascades, e.g. Jak/Stat, NADPH oxidase, mitogen activated protein kinases (MAPK)/p38, extracellular regulated(ERK)-1/2 and c-JunN-terminal kinase (JNK). These phenomena result in the activation of transcription factors, such as nuclear factor (NF-kB) or IFN-stimulated response elements (ISRE) enhancing expression of proinflammatory mediators (modified, based on 8,23). lines 162-167.

Figure 2: Effects of advanced glycation products (AGEs) on adipose tissue dysfunction and the development of obesity-related complications. Lines 513-514

Comments 2 & 3

  • I don’t think the authors can state: “… evidence for anAGE-RAGE pathway involvement in the regulation of adipogenesis” based on reference 31 as this paper only test the osteogenic differentiation. Also, this paper might suggest a positive effect on adipogenesis by AGEs in murine 3T3-L1 preadipocytes (PMID: 23150674), while this paper in human MSCs suggests a negative effect (PMID: 31875018).
  • Similarly, references 33 or 34 do not measure adipocyte differentiation, but only glucose uptake. Consequently, this statement needs revision: “Recent study suggests that accumulation of AGEs in the ECM, independently of RAGE pathway, impairs adipocytes’ differentiation and metabolism [33,34]”. In fact, I think reference 34 might be wrong here.

We thank the Reviewer for drawing attention to these inaccuracies. As suggested, we have revised the references and reworded the paragraphs in question.

Exposure to a high-fat, high-sugar diet negatively impacts on ASCs in adipose and bone tissue. Excessive energy intake promotes AGE formation, which impairs the proliferation and differentiation potential of ASCs, manifested as the decreased cell counting kit-8 (CCK-8) protein level and alkaline phosphatase (ALP) activity. However, the effect can be species-specific. In mice, a high-fat and high-carbohydrate diet leads to a lower expression of osteogenic differentiation genes (Alp, Opn, Ocn, and Runx2) in ACCs. This effect may result from DNA methylation in ASCs and the key role this process plays in the Wnt/β-catenin signaling pathway. Indeed, AGEs drive the downregulation of Wnt signaling molecules (β‐catenin, lymphoid enhancer binding factor 1 (LEF1), and glycogen synthase kinase 3 beta GSK3β ), while enhancing the expression of methyltransferase genes [34]. However, in vitro studies using human mesenchymal stem cells (MSCs) suggest that exposure to AGEs inhibits their adipogenic differentiation (assayed by oil red O staining, lipoprotein lipase production, and intracellular triglyceride content) without incurring significant impairments of osteogenic development [10].

In addition to the evidence for AGE–RAGE pathway involvement in the regulation of ASCs’ fate, there are data on its contribution to adipose tissue expansion via the modulation of adipocyte senescence. For instance, in murine preadipocytes (3T3-L1 cells) activation of the AGE–RAGE axis, probably by blockade of p53 protein, restores the adipogenic potential of these cells. Interestingly under these experimental conditions, AGEs showed no effect on adipogenesis in young preadipocytes and the exact mechanism of this phenomenon remains unknown [35]. Lines 187-207.

Comment 4

4)    The authors should carefully state when conclusions/experiments/references are based on murine, human, or a combination.

For example, “Recent study suggests that accumulation of AGEs in the ECM, independently of RAGE pathway, impaired adipocytes' differentiation and metabolism [33,34]". The conclusion here is based on murine and human adipocytes in reference 33, and the sentence could be "Recent studies suggest that accumulation of AGEs in the ECM, independently of the RAGE pathway, impairs glucose uptake in mouse and human adipocyte models".

We thank the Reviewer for drawing attention to this important aspect. In the original version of the manuscript, information on the experimental model (cell lines, animal models, clinical trials) was included in Table 1. In the revised version of the paper, we have also added this information in the main text (all changes are marked in green). In addition, the paragraph mentioned in the commentary has been rewritten with appropriate literature references.

Regardless of the direct impact on adipocytes, AGE formation in hyperglycemic conditions damages other components of adipose tissue. Recent studies on 3T3-L1 preadipocytes suggest that accumulation of AGEs in the ECM, independently of the RAGE pathway, impairs adipocyte differentiation [36]. In addition, through ECM modification, AGEs have direct effects on cell niches and plasma membranes via loss of their plasticity, and therefore induce alterations in cellular signaling and cytoskeletal organization. Collectively, with AGE accumulation, adipogenesis is reduced, with the downregulated differentiation of fibroblasts and adipocytes [36]. Moreover, the accumulation of AGEs in the ECM also results in impaired glucose uptake and may contribute to the development of insulin resistance, as was shown in human primary adipocytes isolated from diabetic and non-diabetic obese individuals [37]. Lines 208-218.

Comment 5 

  • The adipose tissue is extremely heterogeneous. The authors should elaborate on which specific cells they are discussing, and when it is whole adipose tissue (a mix of all cell types). This also applies to genetic mouse models, when is it constitutive whole-body deletion, when is it constitutive cell-type specific and when is it inducible. This would be important information for the reader to interpret the summary.

Following the reviewer's suggestion, we have edited the manuscript to clarify whether the phenomena described relate to individual cells, tissues or the whole organism and, in the case of genetic studies involving animals, whether knockout/overexpression was generalised or localised. All of the above changes are highlighted in green.

Thus, the global deletion of RAGE in mice protects from this effect, even during exposure to an HFD [24,39]. Experiments on RAGE-/- murine adipocytes strongly supported this evidence, indicating that RAGE inhibits thermogenesis in WAT and BAT via diminishing the protein kinase A (PKA)-mediated phosphorylation of hormone-sensitive lipase (HSL) and other pathways (e.g., p38 mitogen-activated protein kinase, MAPK) involved in the regulation of energy balance [24]. Lines 226-231.

This concept is based on a mouse model, in which the transplantation of adipose tissue from mice with a global overexpression of RAGE to wild-type (WT) mice promoted obesity and insulin resistance [24].

Lines 240-243.

This theory was followed by a series of studies on mice and cell cultures indicating that the downregulation of Toll-like receptor 2 (TLR2) and Toll-like receptor 4 (TLR4) in diet-induced obesity improves adipocyte differentiation and insulin sensitivity in mice [43,44]. However, experiments on mice fed an HFD and murine adipocytes revealed that the suppression of TLR2 enhances adipogenesis; TLR4 results in an opposite effect [44,45]. Lines 250-255.

Using 3T3-L1 cells and mouse models, researchers observed that in hyperglycemic conditions, oxidative stress and the NF-κB pathway mediate ApoE abolishment. Blocking ApoE could partially participate in suppressing adipocyte triglyceride synthesis because the transplantation of wild-type adipocytes in mice with a global ApoE knockout results in less triglyceride accumulation. Lines 264-268.

Animals with a global knockout of the RAGE gene (RAGE−/−) and exposed to an HFD are characterized by an improved glucose tolerance compared with the wild-type (WT) controls reared under the same conditions [50]. Lines 277-279.

Moreover, AGE-stimulated murine adipocyte hypertrophy, which is accompanied by a downregulation of insulin sensitivity genes (e.g., glucose transporter type 4 and adiponectin), may contribute to the diminished glucose uptake and impaired insulin signaling [52]. Lines 292-295.

Consistently, global RAGE knockout correlates with diminished macrophage migration to adipose tissue and lower systemic IL-6 concentrations in mice on an HFD [50]. Perigonadal adipose tissue (PGAT) and bone marrow derived from RAGE-/- mice on an HFD exhibit a reduced expression of inflammatory markers typical for M1 macrophages compared with controls. In addition, macrophages in PGAT of RAGE-/- mice were shown to express lower levels of CD11c marker which, together with polarization into the anti-inflammatory M2 phenotype, result in the upgraded regulation of insulin secretion [53,54]. Lines 297-304

In mice with obesity-associated diabetes, leptin downregulation leads to an overexpression of RAGE in β-cells, and therefore inhibits insulin infusion and mediates AGE-elicited pancreatic islet apoptosis [63]. Lines 341-343.

AGEs with leptin inhibitory properties include glycolaldehyde-modified BSA (that binds RAGE) and oxidized low-density lipoproteins which, in turn, interact with CD36 receptor, as shown in 3T3-L1 adipocytes and mouse epididymal adipocytes [64]. Lines 346-349

Proper PPAR-γ expression in murine adipose tissue hampers AGE formation and acts as a downregulator of RAGE, thus contributing to the prevention from AGE–RAGE axis-mediated cardiovascular disorders [66]. Lines 353-355.

One study revealed that the stimulation of RAGE by its agonist N(ε)-(carboxymethyl)lysine (CML) resulted in a significant increase in inflammatory molecules expression (such as plasminogen activator inhibitor (PAI)-1 and IL-6) with a simultaneous reduction in adiponectin secretion by human preadipocytes [59]. Lines 368-371.

Indeed, Maeda et al. reported that epithelium-derived factor (PEDF), which is a multifunctional, antioxidant glycoprotein, inhibits the AGE–RAGE-induced suppression of adiponectin mRNA level in human visceral adipocytes through the suppression of nicotinamide adenine dinucleotide phosphate (NADPH) oxidase [70]. Lines 373-377.

Comment 6

6)    Further, I'm missing general information about which cells in the adipose tissue have RAGE receptors. This information should be possible to find in publicly available single-cell/nuclei datasets. Further, is it clear which cells in the adipose tissue secrete AGE ligands, or are most AGEs exogenous?

In response to the Reviewer comment, the following paragraphs have been added:

Both endogenous and exogenous AGEs have the ability to bind to a specific RAGE receptor present on the surface of, among others, endothelial cells, muscle cells, immunocompetent cells, glomerular podocytes, and neurons. RAGE expression has been found in many adipose-tissue-forming cells: adipocytes, their precursors, stromal cells, vasculature, or infiltrating macrophages [24]. Lines 140-144

AGEs present in adipose tissue can be either exogenous (supplied by the circulation) or endogenous (formed locally as a result of adipose-tissue-dysfunction-related oxidative stress or diabetes-related hyperglycemia). Lines 110-112.

Reviewer 5 Report

The manuscript is interesting and the authors propose a current view about the participation of RAGE in adipose tissue metabolism.  Manuscript is well write, structure is coherent and the authors present and discuss principal aspects related with the proposal of the review.  

Nonetheless, I have some minor comments.

I. Minor comments:

1. Improve redaction of the aim the manuscript.

2. I suggest discuss briefly the participation of oxidative stress in adipose dysfunction tissue in obesity. PMID: 33026007

3. It is necessary to add figures, for example a figure that show principle molecular pathways. Considering all topics, the manuscript can be have two figures.     

Manuscript is well write, but in necessary correct some redaction mistakes.  

Author Response

Reviewer 5

The manuscript is interesting and the authors propose a current view about the participation of RAGE in adipose tissue metabolism.  The manuscript is well write, structure is coherent and the authors present and discuss principal aspects related with the proposal of the review. 

Nonetheless, I have some minor comments.

We would like to thank the reviewer for the positive reception of our work and the comments, which will undoubtedly improve the quality of the manuscript.

Minor comments:

Comment 1

  1. Improve redaction of the aim the manuscript.

As suggested by the Reviewer, the part of the introduction concerning the purpose of the work has been rewritten. In addition, the manuscript has been sent to a professional English-language editor (please see attached certificate).

The aim of this narrative review is to summarize the data on the role of the RAGE pathway in adipose tissue dysfunction in obesity and its metabolic complications. First, the mechanisms of AGE formation and the RAGE signaling pathway are briefly presented. Next, we outline the role of the RAGE pathway in the regulation of adipogenesis and adipose tissue function. We then summarize data from animal and human studies on the involvement of the RAGE pathway in obesity, diabetes, and cardiovascular disease. Finally, we discuss therapeutic perspectives based on interference with the RAGE pathway. Lines 84-91.

Comment 2

  1. I suggest discuss briefly the participation of oxidative stress in adipose dysfunction tissue in obesity. PMID: 33026007

As suggested by the Reviewer, the relevant paragraph and the reference on the role of oxidative stress in the development of obesity-related adipose tissue dysfunction have been added to the Introduction.

Consequently, the dysfunction of adipose tissue impacts homeostasis of the entire body. This situation takes place in the course of obesity, when excess nutrients accumulated in adipocytes lead to mitochondrial dysfunction and associated oxidative stress [2]. In turn, the oxidative stress can increase preadipocyte proliferation, adipocyte differentiation, and the size of mature adipocytes. In obesity, oxidative stress is not limited to adipose, and the reactive oxygen species (ROS) can impair function of hypothalamic neurons that control satiety and hunger behavior, in a vicious circle mechanism [3]. Lines 44-50.

Comment 3

It is necessary to add figures, for example a figure that show principle molecular pathways. Considering all topics, the manuscript can be have two figures.     

Following the Reviewer's valuable comment, we have added two figures to the manuscript: Figure 1 on the molecular pathways involved in the AGE receptor and Figure 2 summarising the potential effects of overexpression and activation of this receptor in adipose tissue.

Figure 1. A simplified scheme of RAGE-mediated signaling. The interaction between the advanced glycation end products (AGEs) and their receptor (RAGE) stimulates several intracellular signaling cascades, e.g. Jak/Stat, NADPH oxidase, mitogen activated protein kinases (MAPK)/p38, extracellular regulated(ERK)-1/2 and c-JunN-terminal kinase (JNK). These phenomena result in the activation of transcription factors, such as nuclear factor (NF-kB) or IFN-stimulated response elements (ISRE) enhancing expression of proinflammatory mediators (modified, based on 8,23). Lines 162-167.

Figure 2: Effects of advanced glycation products (AGEs) on adipose tissue dysfunction and the development of obesity-related complications. Lines 513-514.

Round 2

Reviewer 4 Report

I think the authors response is satisfying and the language has been improved.